# Heat Transfer Correlations for Smooth and Rough Airfoils

Sepehr Samadani [†] and François Morency *,[†]

Thermo-Fluids for Transport Laboratory (TFT), Department of Mechanical Engineering, École de Technologie Supérieure (ÉTS), 1100 Notre-Dame St W, Montréal, QC H3C 1K3, Canada

* Correspondence: francois.morency@etsmtl.ca

† These authors contributed equally to this work.

**Abstract:** Low-fidelity methods such as the Blade Element Momentum Theory frequently provide rotor aerodynamic performances. However, these methods must be coupled to databases or correlations to compute heat transfer. The literature lacks correlations to compute the average heat transfer around airfoil. The present study develops correlations for an average heat transfer over smooth and rough airfoil. The correlation coefficients were obtained from a CFD database using RANS equations and the Spalart–Allmaras turbulent model. This work studies the NACA 0009, NACA 0012, and NACA 0015 with and without the leading roughness representative of a small ice accretion. The numerical results are validated against lift and drag coefficients from the literature. The heat transfer at the stagnation point compares well with the experimental results. The database indicates a negligible dependency on airfoil thickness. The work presents two correlations from the database analysis: one for the smooth airfoils and one for the rough airfoils. For the zero lift coefficient, the average Nusselt number is maximum. This increases with $Re^{0.636}$ for the smooth surface and with $Re^{0.85}$ for the rough surface. As the lift increases, the average Nusselt is reduced by values proportional to the square of the lift coefficient for the smooth surface, while it is reduced by values proportional to $Re$ and the square of the lift coefficient for the rough surface.

**Keywords:** heat transfer; roughness; symmetric airfoil; Nusselt number; RANS equations; Spalart–Allmaras turbulence model

## 1. Introduction

Aircraft and helicopters need ice protection systems (IPSs), such as anti-icing or de-icing systems, to safely fly through supercooled water droplets in clouds [1]. Thermal melting is the most common method of preventing ice buildup on critical surfaces. The main goal for IPSs is to use the lowest power while ensuring safe ice removal for various atmospheric conditions [2]. The IPS should remove ice accretion to avoid aerodynamic performance degradation without exceeding the critical temperature of the materials, even if the IPS operates outside in-flight icing conditions. Experimental methods [3,4] or computational fluid dynamics (CFD) methods [5–7], coupled with reduced-order modeling and optimization methods provide insightful information for the detailed design. However, these high-fidelity tools are expensive to use in the early development phases when several aircraft or helicopter configurations are evaluated. A medium fidelity method such as the vortex lattice method, (VLM) coupled with a 2D viscous database, is computationally less expensive and enables aircraft icing studies [8].

Researchers have coupled low-fidelity methods, such as Blade Element Momentum Theory (BEMT), or medium fidelity methods, such as the VLM, and airfoil lookup tables or correlations to estimate aerodynamic forces, performance degradation or heat transfer. VLM and a drag correlation based on airfoil experimental data enable the computationally efficient calculation of the vibration of helicopter rotors [9]. The non-linear unsteady VLM coupled to 2D RANS or empirical databases adequately predicts helicopter rotor aerodynamics in hover [10]. Nonlinear VLM coupled to 2.5D RANS

sectional data allows the prediction of the maximum lift coefficient for a 3D swept wing [11]. The results agree reasonably well with 3D RANS solution, and the proposed approach is well suited for preliminary design. A model for predicting the torque for a rotor under icing conditions is proposed based on the BEMT [12,13] and a correlation for airfoil performance degradation in an icing environment [14]. Samad et al. [15] coupled the VLM with a 2D RANS database to compute lift and drag. The effective angle of attack and the Reynolds number are determined at each radial position of the blade. Then, the average heat transfer coefficients are computed using a correlation based on a RANS database for an airfoil under fully turbulent conditions [16].

The correlations for the aerodynamic forces on 2D airfoil often related the lift coefficient, $c_l$, to the angle of attack $\alpha$, and the drag coefficient $c_d$ to $c_l$. For a typical airfoil at high Reynolds numbers, $c_l$ linearly evolves with $\alpha$ until the flow separates from the surface, close to the stall angle [17]. The maximum lift coefficient and the aerodynamic forces after the stall angle depend on many parameters, the Reynolds number and the airfoil shape included among them [18]. Researchers used experimental and RANS databases to build correlation for both $c_l$ and $c_d$ at the post-stall angle [19]. Before the stall angle, Hoerner [20] suggests that $c_d$ is a function of the square of the lift coefficient. Usually, $c_d$ has a minimum value $c_{d,min}$ for $-2° < \alpha < 2°$ [21]. At this minimum drag value, the lift coefficient is $c_{l,mind}$, such that

$$c_d = c_{d,min} + A(c_l - c_{l,mind})^2$$

Recent research has suggested a novel estimation method for $c_{d,min}$ over smooth airfoil based on an extensive RANS database, including 40 airfoil shapes [22]. Gotten et al. decomposed $c_{d,min}$ in the friction drag and the pressure drag, both functions of the airfoil shape parameters. This estimation method extends the Reynolds range of the validity of the previous correlations.

Most correlations relate the heat transfer to the Reynolds number, $Re$, and the Prandtl number, $Pr$. For well-studied geometries such as a flat plate, the Nusselt number $Nu$ is related to $Re$ and $Pr$ [23]. Separate correlations model the laminar or turbulent flows, but Lienhard [24] suggested approximations that also include the transitional flows. For a cylinder in cross flow, the $Nu$ [25] or the Frossling number $Fr = Nu/\sqrt{Re}$ [26] correlate with $Re$ and $Pr$. For ice-roughened surfaces, limited data are available, but some researchers use $Fr$ instead of $Nu$ [27].

The literature suggests heat transfer correlations for heat exchanger analysis. In particular, recent studies used vortex generators, geometries that share some analogy with wings, to enhance the heat transfer in tubes. The Dittus–Boelter equation, $Nu = 0.023Re^{0.8}Pr^{0.4}$, is modified to consider the geometry, the angle of attack, and the spacing of the vortex generators. The correlation coefficients are obtained using the least-square regression method with experimental data [28,29]. Heat transfer enhancement by vortex generators was also numerically studied using CFD [30] data. Instead of correlations, artificial neural network models have been used for the performance prediction and optimization of complex heat exchanger geometries [31].

Compared to flat plates and cylinders, heat transfer correlations for airfoil have a minimum of two additional parameters: the airfoil shape and the angle of attack, $\alpha$. Fewer experimental data and correlations are publicly available for the convective heat transfer around airfoil. Most notably, the works of [32,33] studied the heat transfer coefficient in the leading edge area of an NACA0012 with smooth and rough surfaces, for $-6° < \alpha < 8°$. The local $Fr$ over the smooth surface was independent of $Re$ close to the stagnation point. The rough surfaces consisted of 2 mm diameter hemispheres arranged in four patterns. Dukhan et al. experimentally measured the $Fr$ in the first 10% of an NACA 00012 with mildly rough glaze and rough glaze ice with horns [34]. At $\alpha = 0°$, the $Fr$ at stagnation point is a quadratic function of $Re$. Downstream from the stagnation point, the local $Fr$ is a polynomial function of the distance along the airfoil. The polynomial constant values vary with $Re$. When ice accretes on an airfoil, the local $Fr$ increases in time as the ice roughness grows [35]. Average heat transfer correlations

for NACA0010 [36] and the NACA 63-421 [37] are proposed based on experimental results. Both correlations relate the average Nusselt $\overline{Nu}$ to $Re$. Further works expand the correlation for NACA 63-421 to include the effects of $\alpha$ on $\overline{Nu}$ based on the experimental measurements between $0° < \alpha < 25°$ [38].

Instead of experiments, Samad et al. [15] used CFD to build a correlation for the $Fr$ number as a function of $Re$ and $\alpha$ for the NACA 0012 under fully turbulent flow conditions. They further improved the correlation to predict the post stall heat transfer using a cubic variation with $\alpha$ [16]. Using experimental results on a rotor, Ref. [39] proposed correlations to include the effects of water spray and the presence of an ice layer for $\alpha = 6°$.

The design of IPS requires the heat loss by the airfoil to the cold airflow. The BEMT and the unsteady VLM enable the quick prediction of the aerodynamic forces and the average heat transfer if they are coupled with correlations to include viscous effects. However, correlations for average heat transfer are only available for three airfoils with smooth surfaces at $Re < 3 \times 10^6$. The present study partially addresses this gap by studying the effects of the airfoil thickness, the lift coefficient and the roughness on symmetric airfoil for $Re < 6 \times 10^6$. The study develops correlations for the average heat transfer over smooth and rough airfoil. A RANS database for aerodynamic flows over symmetric airfoils helps build the correlations. The study focuses on heat transfer before the stall for fully turbulent flow in the range $0.625 \times 10^6 \leq Re \leq 6.0 \times 10^6$. Three airfoil thicknesses to cord ratios, $t/c = 0.09$, 0.12, and 0.15, assess the result sensitivity to the thickness. The addition of leading edge roughness models the ice accretion effects. First, the paper presents the RANS model used together with the meshes and the proposed average heat transfer function. Second, the lift, drag, and heat transfer from the literature validate the RANS model predictions. Then, the database is verified and used to suggest two heat transfer correlations: one for the smooth airfoils, and one for the rough airfoils.

## 2. Materials and Methods

In the context of rotor aerodynamics, the work assumes that each blade section acts as a quasi-2D airfoil to produce aerodynamic forces and heat transfer, and thus 3D effects such as wing tip vortices are not included. The flow of air is compressible and fully turbulent, and no laminar boundary layer region exists on airfoil. The heat transfer by radiation is neglected. The conduction and the thermal inertia in the airfoil skin are negligible as the temperature is constant in both time and space.

The heat transfer coefficients over three airfoils were obtained using CFD. The airfoils were selected to study the results' sensitivity to the thickness-to-cord ratio. The symmetric NACA 0009, NACA 0012, and NACA 0015 have a maximum thickness-to-cord ratio, $t/c$, of 0.09, 0.12 and 0.15, respectively, located at $x/c = 0.3$, as shown in Figure 1. The surface is either smooth or rough to model the effects of small ice accretion. The roughness covers the leading edge region, in the first 8% of the cord. The ice accretion depends on atmospheric conditions and can sometimes extend up to 15% of the cord. The present choice of the first 8% follows from previous experimental works [40]. For this study, $\alpha$ goes from $0°$ up to the stall angle.

The compressible RANS equations model the airflow around the airfoil [41]. The air is considered to be a perfect gas, with a specific gas constant $R = 287.058 \, J/kgK$ and specific heat ratio coefficients at a constant pressure and volume $\gamma = c_p/c_v = 1.4$. The Sutherland formula gives the dynamic viscosity $\mu$ for air as a function of temperature $T$

$$\mu = \frac{1.45 \times 10^{-5} T^{3/2}}{T + 110}. \tag{1}$$

The thermal conductivity $k$ is obtained from $Pr$

$$k = \frac{c_p \mu}{Pr}. \tag{2}$$

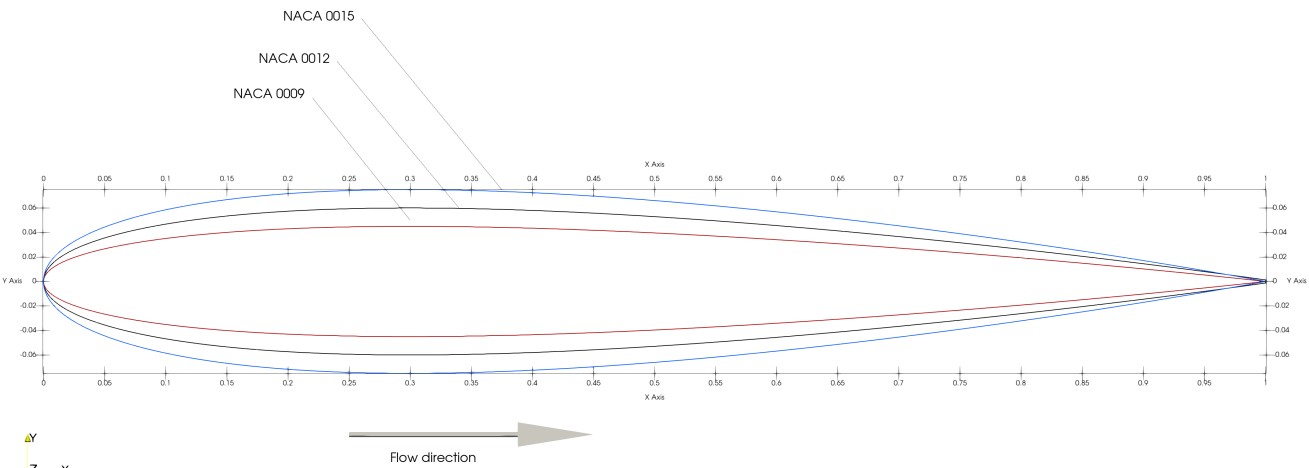

**Figure 1.** Comparison of the three airfoil geometries for a cord $c = 1$ m. Maximum thickness ratios $t/c$ range from 0.09 to 0.15.

The Prandtl number has a constant value of 0.72. The effects of relative humidity on the air properties are neglected [42].

The flow is fully turbulent, for both smooth and rough airfoils. The Spalart–Allmaras turbulent model (SA) is used for the smooth surface [43]. The SA model is commonly used in aeronautics to model attached flow as it gives satisfactory predictions for lift and drag [41]. The Boeing method corrects the SA model (SA-rough) to predict the flow over rough walls [44]. This model accounts for roughness effects on the wall shear stress with the equivalent sand grain roughness, $h_s$, but an additional correction is needed for the heat transfer prediction. The two parameters of the Prandtl correction model [45] add the function $\Delta Pr_t$ to the otherwise constant turbulent Prandtl number, $Pr_t = 0.9$. Briefly, the correction $\Delta Pr_t$ depends on $h_s$ and the physical roughness height $h$. For a roughness Reynolds number $Re_s$ above 70, the correction is

$$\Delta Pr_t = 0.136 \frac{Re_s^{0.45} Pr^{0.8}}{1.92}, \tag{3}$$

$$Re_s = \frac{\rho u_\tau h_s}{\mu}, \tag{4}$$

$$u_\tau = \sqrt{\tau_w / \rho}. \tag{5}$$

At the wall, the SA-rough model predicts a non-zero eddy viscosity $\mu_t$, and therefore the heat flux at the wall, $q_w$, is

$$q_w = -(k + k_t)\left(\frac{\partial T}{\partial y}\right)_w, \tag{6}$$

$$k_t = \frac{c_p \mu_t}{Pr_t + \Delta Pr_t}. \tag{7}$$

where $\left(\frac{\partial T}{\partial y}\right)_w$ is the temperature gradient at the wall. Over smooth surfaces, the SA model imposes $\mu_t = 0$ and $\Delta Pr_t = 0$.

The local Nusselt number is defined based on the airfoil cord $c = 1$ m and the recovery temperature at freestream $T_t$

$$Nu = \frac{q_w c}{(T_w - T_t)}, \tag{8}$$

$$T_t = T_\infty (1 + Pr^{1/3} 0.5(\gamma - 1) Ma^2). \tag{9}$$

where $Ma$ and $T_\infty$ are the farfield Mach number and temperature. An average Nusselt number $\overline{Nu}$ over the airfoil wet surface $s$ can also be defined, such that

$$\overline{Q} = \frac{\int q_w ds}{s}, \tag{10}$$

$$\overline{Nu} = \frac{\overline{Q}c}{(T_w - T_t)}, \tag{11}$$

for a constant wall temperature $T_w = 300\,K$.

The 2D computational domain consists of a circle of radius $100c$ with the airfoil leading edge located at the center, as illustrated in Figure 2. Riemann boundary conditions are imposed at the farfield boundary. The Mach number is $Ma = 0.2$ and the temperature is $T_\infty = 268.15\,K$. The static pressure at farfield $P_\infty$ set the density $\rho = P_\infty / RT_\infty$ and the Reynolds numbers, $Re = \rho V_\infty c / \mu$. The turbulence model variable $\tilde{v}$ is set to three times the kinematic viscosity at farfield. At the airfoil wall, a no-slip boundary condition is imposed together with a constant temperature. For the rough leading edge, the standard leading-edge roughness consists of carborundum grains applied to the surface of the model [40]. In [40], $0.279\,mm$ carborundum grains are applied to a $c = 0.6096\,m$ airfoil. The equivalent sand grain roughness is $h_s/c = 0.001$ and the roughness height is $h/c = 0.00458$, corresponding to the small ice accretions between deicing cycles.

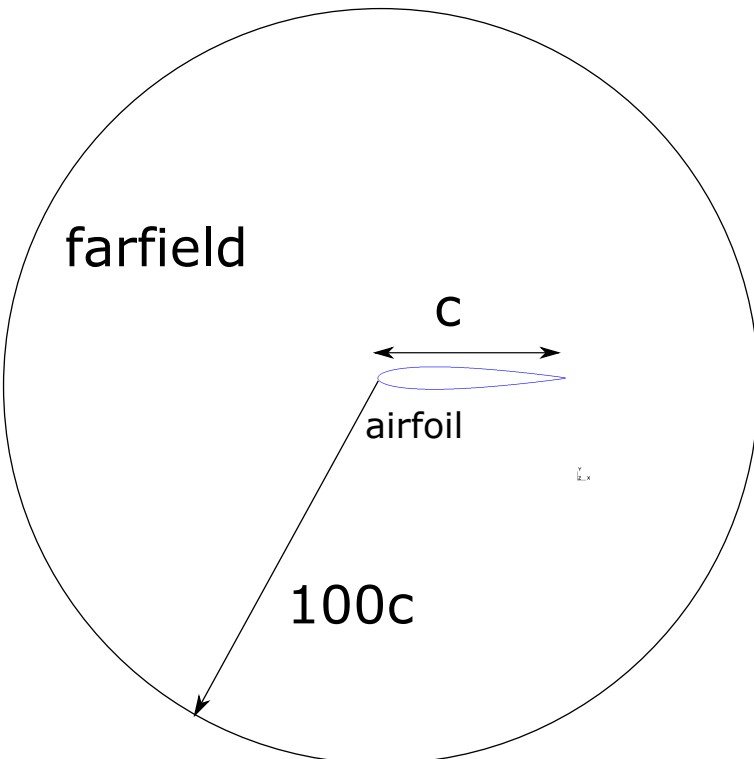

**Figure 2.** Computational domain around airfoil, but the airfoil is out of scale. The farfield boundary is a circle of the radius of $100c$ centered around the airfoil leading edge.

Mixed meshes discretize the computational domain. The mesh generator Gmsh [46] builds rectangular elements with a growth ratio of 1.15 for a normal surface distance lower than $0.02c$ and triangular elements farther away. For each airfoil, three meshes are constructed to enable a grid convergence study. Figure 3 shows a close-up view of the airfoil leading edge for the NACA 0012 medium mesh. For the coarse, medium and fine meshes, the first node above the surface is located at $3.5 \times 10^{-6}\,m$, $2.5 \times 10^{-6}\,m$ and $1.8 \times 10^{-6}\,m$. At $Re = 6 \times 10^6$, this corresponds to maximum dimensionless wall distances $y^+ = 0.48, 0.35$, and $0.25$ above the smooth surface, respectively. For the rough surface and the medium mesh, $y^+ = 0.7$. The number of nodes on the airfoil surface is approximately

400, 560, and 780 with a smaller node spacing close to the leading edge and the trailing edge. For the medium mesh in Figure 3, at $Re = 6 \times 10^6$, $x^+ \approx 70$ is close to the leading edge.

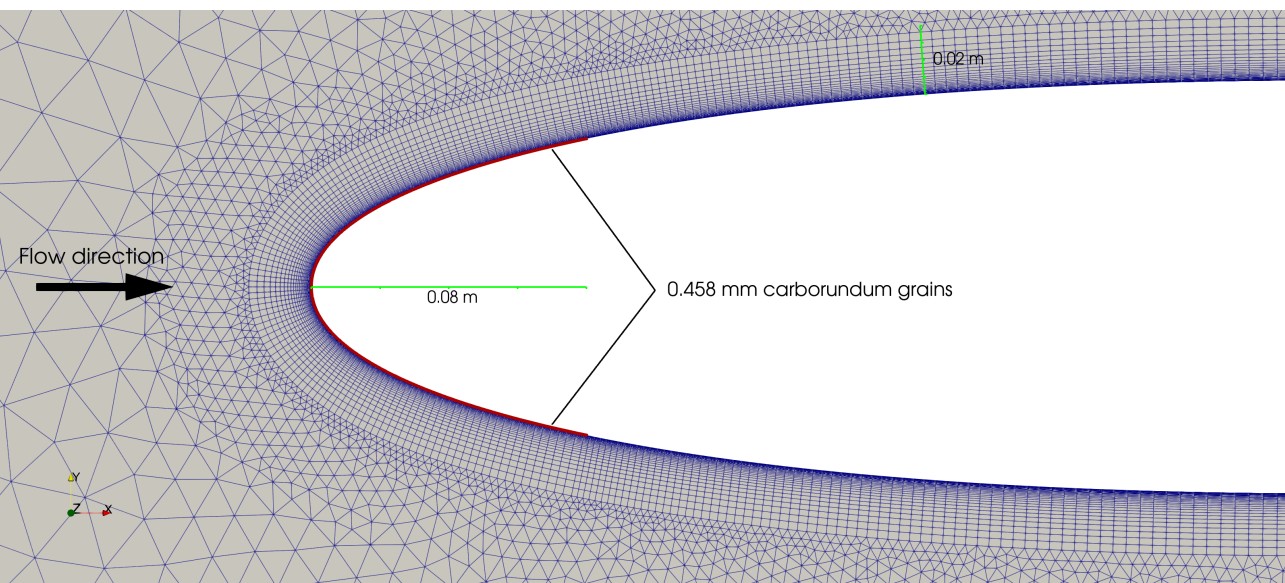

**Figure 3.** Medium mesh close to the leading area for the NACA 0012. When activated, the rough surface covers the first $0.08c$.

The steady RANS equations are solved with a modified version of the finite volume code SU2 version 6.2 [47]. The modifications implement the SA-rough model and the turbulent Prandtl number correction for flow over the rough surface, as validated in [45]. The RANS and SA equations are discretized over the mesh using a cell-vertex scheme with median dual control volumes. The space numerical integration uses the Roe scheme for the RANS inviscid terms [48]. To reach second-order accuracy, the MUSCL scheme is used with a Venkatakrishnan–Wang slope-limiting method [49]. For the viscous terms, the gradients are computed using the Green–Gauss theorem [41]. The steady-state solution is iteratively reached using an implicit time stepping scheme. An Euler implicit time integration method is used for the flow equations with an adaptive Courant–Friedrichs–Lewy (CFL) number, with $0.5 < CFL < 10$. The flexible generalized minimal residual method, FGMRES, solves the resulting linear system of equations using a LU-SGS preconditioning [47]. The iterations stop when the residual of the density equation is below $10^{-11}$.

The grid convergence method (GCI) is applied [50] to study the effects of the grid refinement on $\overline{Nu}$. The maximum GCI for the three smooth airfoils at $\alpha = 0°$, $10°$, and $15°$ is 0.18%. For the three rough airfoils, at $\alpha = 0°$, $5°$, and $8°$, the maximum GCI is 2%. This is higher than for the smooth airfoil, but still acceptable to build a database for correlations. The GCi is a measure of the grid-induced errors. At 2%, it is lower than the error induced by the turbulence model choice between $\approx$ 3% and 5% according to [16].

This work assumes that $\overline{Nu}$ is a function of $Re$, $c_l$, and $Pr$

$$\overline{Nu} = (A Re^B + C\, c_l^2) Pr^{1/3} \tag{12}$$

for the attached and mildly separated flow. The $Pr$ dependency is assumed for compatibility with previous correlations.

The database contains 360 CFD simulations. The simulations are run at five Reynolds numbers $Re = 0.625 \times 10^6$, $1.25 \times 10^6$, $2.5 \times 10^6$, $4.0 \times 10^6$, $5.0 \times 10^6$, and $6.0 \times 10^6$. The $\alpha$ range is limited by the stall angle, reached at lower values for the rough airfoils. For the smooth airfoils, $\alpha = 0°$, $2°$, $4°$, $6°$, $8°$, $10°$, $12°$, $13°$, $14°$, and $15°$. For the rough airfoils, $\alpha = 0°$, $2°$, $4°$, $6°$, $8°$, $9°$, $10°$, $11°$, $12°$, and $13°$. The correlation coefficients $A$, $B$, and $C$ are not sensitive to the three thickness ratios $t/c$ selected.

## 3. Results

The coefficients *A*, *B*, and *C* are determined by fitting the nonlinear regression model Equation (12) to the CFD dataset [51]. The CFD results are first validated against experimental lift and drag coefficients for the three airfoils. The Nusselt number predictions are validated against experimental data for the NACA 0012. Then, the heat transfer results are verified at $\alpha = 0°$ over the Reynolds range and at $Re = 6 \times 10^6$ over the $\alpha$ range. Finally, $\overline{Nu}$ is correlated to $Re$ and $c_l$ for smooth and rough surfaces.

### 3.1. Validation

With a maximum GCI around 2%, the medium mesh precision is sufficient to build average heat transfer correlations. The CFD results for the NACA 0009 and NACA 0012 on the medium mesh are validated against experimental lift and drag coefficients from [40,52]. Although the compressible RANS equations are solved, the compressible effects are essentially negligible. The CFD Mach number is 0.2 and the Reynolds number is $Re = 6 \times 10^6$. Figure 4 shows the $c_l$ evolution with $\alpha$. For the NACA 0012, two sets of experimental data are plotted. The $c_l$ linearly increases with $\alpha$ for both the NACA 0009 and NACA 0012 until around $\alpha = 12°$. At $\alpha = 13°$, the experimental results for the NACA 0009 show a decrease in $c_l$, whereas the $c_l$ predicted by the SA model keeps increasing, but not linearly. For the NACA 0012 experimental results, the maximum lift coefficient is reached around $\alpha \approx 16°$, above the maximum $\alpha = 15°$ for the CFD simulation.

Figure 4 also compares the $c_d$ as a function of $c_l$. For NACA 0012, $c_d$ from [52] experiments, obtained by tripping the boundary layer at the airfoil leading, is closer to the CFD results than the untripped Abott results for $c_l < 1$. The fully turbulent CFD results for both airfoils are close until $c_l \approx 1.2$. Then, the predicted $c_d$ increases faster for the NACA 0009, most probably because the stall occurs at $\alpha = 13°$. Note that Abbott results do not include $c_d$ values above the stall angle.

For the NACA 0015 geometry, the SA results are compared to the experimental results [53] and numerical results [54] in Figure 5. The Reynolds number is $Re = 1.6 \times 10^6$. For the SA model, the Mach number is kept at 0.2. The SA model results agree well with the SST turbulence model results. Both numerical results fail to predict the stall angle and maximum lift coefficient, but closely follow the experiments for $\alpha < 12°$. Although the experimental results are for untripped flow, the predicted drag coefficients $c_d$ in Figure 5 are only slightly above the measurements before the stall angle, $\alpha < 15°$. The SST and SA results provide similar drag predictions.

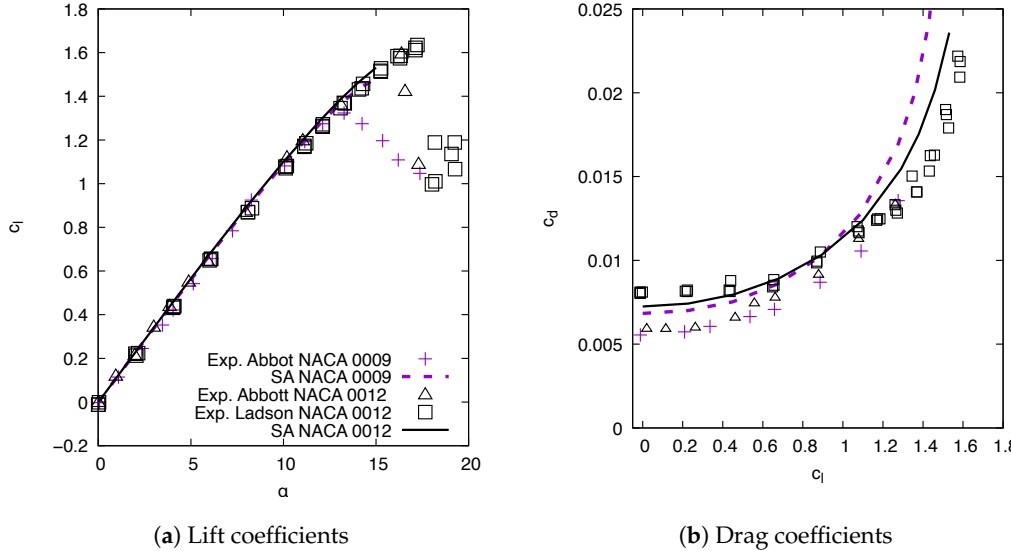

(**a**) Lift coefficients        (**b**) Drag coefficients

**Figure 4.** Comparison between CFD results and experiments for NACA 0009 and NACA 0012 at $Re = 6 \times 10^6$. Legend applies to (**a**,**b**). Ladson results are obtained by tripping the boundary layer at the airfoil leading edge. Abbot results are not tripped.

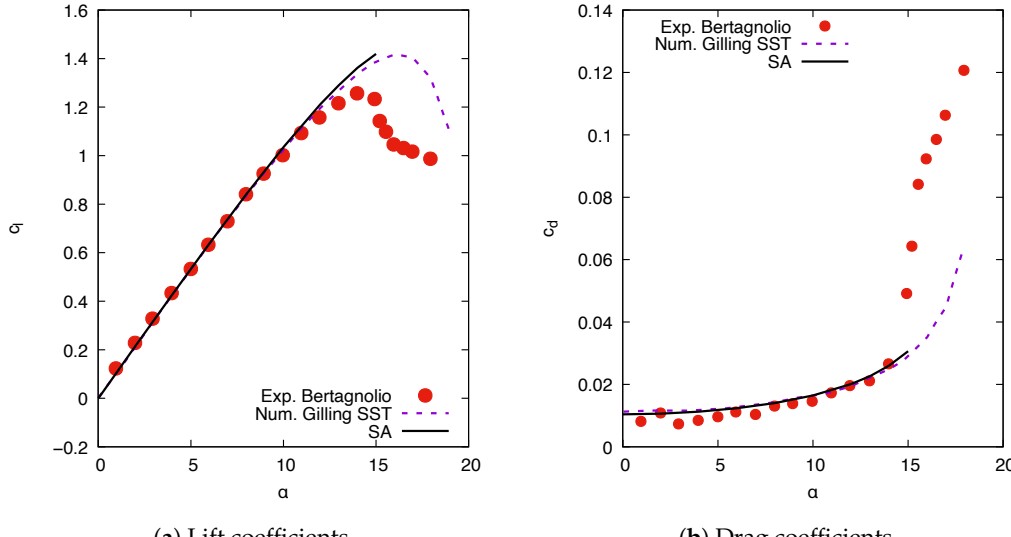

(**a**) Lift coefficients

(**b**) Drag coefficients

**Figure 5.** Lift and drag coefficients comparison between CFD results and experiments for the NACA 0015 at $Re = 1.6 \times 10^6$. Legend applies to (**a**,**b**).

For the rough results, the equivalent sand grain roughness is $h_s/c = 0.001$ and the roughness height is $h/c = 0.00458$, approximately corresponding to the experimental grit 60 size roughness used on the NACA 0012 airfoil of $c = 0.6096$ m [40]. Figure 6a,b compare the predicted $c_l$ and $c_d$ with the experimental results of [40] for the NACA 0012 with roughness at $Re = 6 \times 10^6$. The results for the smooth NACA 0012 are also plotted to ease the comparison. The reduction in the maximum lift coefficient is well predicted by the SA-rough model. The increase in the drag coefficient is similar to that observed experimentally, with a maximum discrepancy of approximately 3%.

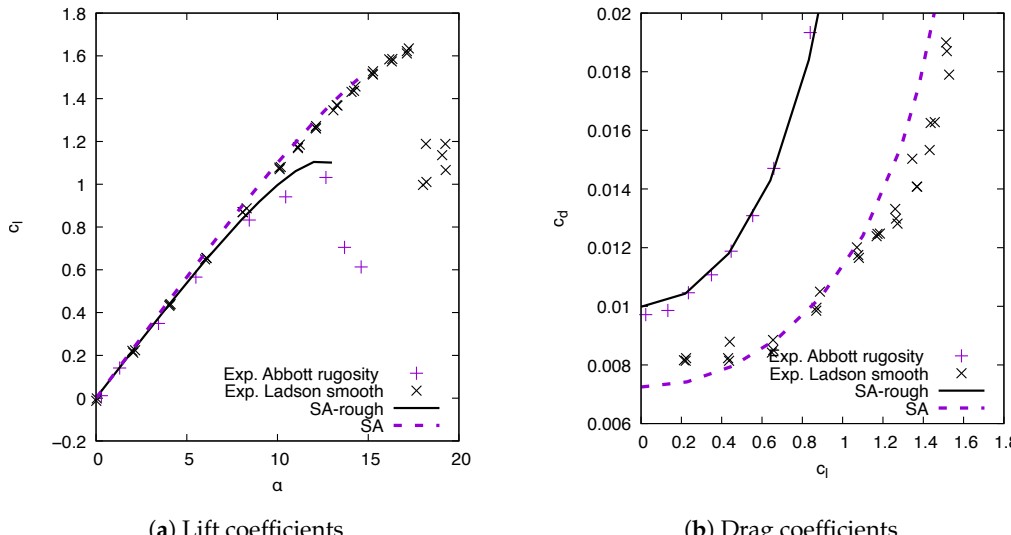

(**a**) Lift coefficients

(**b**) Drag coefficients

**Figure 6.** Lift and drag coefficients comparison between CFD results and experiments for the NACA 0012 with a leading edge roughness at $Re = 6 \times 10^6$.

Almost no data are available for the validation of the CFD average heat transfer prediction over an airfoil. The SA-rough model coupled with the thermal Prandtl correction was previously validated against flat plate results [45], or flow with curvature [55]. No experimental data for average heat transfer over the airfoils at high Reynolds number and fully turbulent flow are available. However, the heat transfer at the NACA 0012 stagnation point is compared with the experiments. Figure 7 shows the predicted heat transfer for both the SA and SA-rough model at the leading edge as a function of $Re$. The angle of attack is

$\alpha = 0$. In [56], a correlation based on the heat transfer measurement at the airfoil leading edge in the IRT wind tunnel is given, as $Nu = 6.818Re^{0.472}$. The correlation is valid for smooth surfaces, and for the Reynolds range of the experiment, $1.2 \times 10^6 < Re < 4.5 \times 10^6$. The SA model prediction closely follows the experimental results. The SA-rough results are also plotted to show that $Nu$ increases with surface roughness at a lower Reynolds number, approximately 8% of increases at $Re = 2.5 \times 10^6$. For $Re \geq 4 \times 10^6$, the heat transfer at the leading edge is unaffected by the roughness.

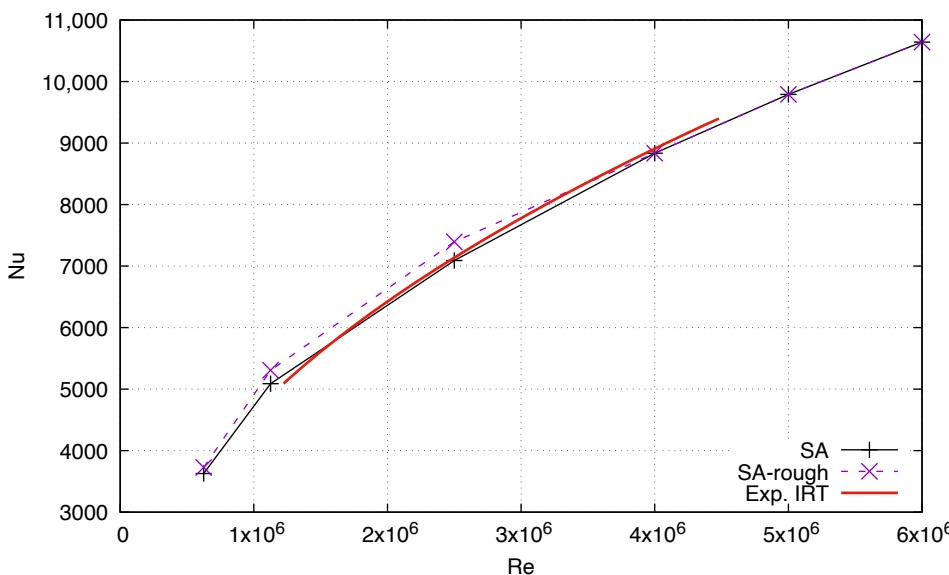

**Figure 7.** $Nu$ at the leading edge: comparison between CFD results and experimental correlation $Nu = 6.818Re^{0.472}$ for the NACA 0012 at $\alpha = 0$.

*3.2. Database Verification*

Previous experiments with airfoils at a lower $Re$ [36,37] and correlations for heat transfer in the turbulent boundary layer [23] predict that $\overline{Nu}$ should increase with $Re$. Figure 8 shows that $\overline{Nu}$ has a function of $Re$ for the three airfoils. Although the roughness height $h_s$ is constant, the roughness Reynolds number, $Re_s$ increases from $\approx 50$ at $Re = 0.625 \times 10^6$, to $\approx 500$ at $Re = 6 \times 10^6$. According to [57], the roughness regime is transitionally rough for $Re = 0.625 \times 10^6$ and fully rough for all the others $Re$.

At $\alpha = 0°$, the results for rough airfoils increase faster than the results for smooth airfoils. The effects of airfoil thickness $t/c = 9\%$, 12%, and 15% on $\overline{Nu}$ are small once the total heat flux is divided by the respective wet surface $s = 2.02c$, $2.04c$, and $2.06c$. For comparisons, the correlation for the turbulent heat transfer over a flat plate of length $c$

$$\overline{Nu} = (0.037Re^{0.8} - 871)Pr^{0.333}$$

is also plotted [23]. The curve for the flat plate gets closer to the smooth airfoil results as $Re$ increases. The discrepancy between the rough and smooth results increases with $Re$, reaching a maximum value of $\approx 9\%$ for $Re = 6 \times 10^6$. This indicates that the Reynolds exponent in Equation (12) must be different for rough and smooth surfaces.

The heat transfer locally increases above the roughness and reduces to smooth values downstream of the first $0.08c$. The fraction of the surface with roughness, $s_r$, changes with increasing airfoil thickness, such that $s_r = 0.089s$, $0.094s$, and $0.099s$. $\overline{Nu}$ blurs the effects of roughness. To emphasize the roughness effects, Figure 9 shows that the Nusselt number averaged over the first $0.08c$

$$\overline{Q_r} \quad = \quad \frac{\int q_w ds}{s_r}, \tag{13}$$

$$\overline{Nu_r} \quad = \quad \frac{\overline{Q_r}c}{(T_w - T_t)}. \tag{14}$$

$\overline{Nu_r}$ is significantly higher in the leading edge area than for the complete airfoil, with a maximum value for the rough surface of $2.0 \times 10^5$ compared to $0.9 \times 10^5$. The results are more sensitive to the roughness than to the airfoil thickness. As in Figure 8, the discrepancy between the smooth and rough results increase with $Re$. At $Re = 0.625 \times 10^6$, $\overline{Nu_r}$ is $\approx 13\%$ higher for the rough surface. At $Re = 6 \times 10^6$, $\overline{Nu_r}$ is $\approx 80\%$ higher for the rough surface. This confirms the need for a different correlation for the flow over the rough surface. The heat transfer decreases as the airfoil thickness increases. However, the effect of the airfoil thickness at $\alpha = 0°$ is negligible, being at least an order of magnitude smaller than the effects of $Re$ and roughness.

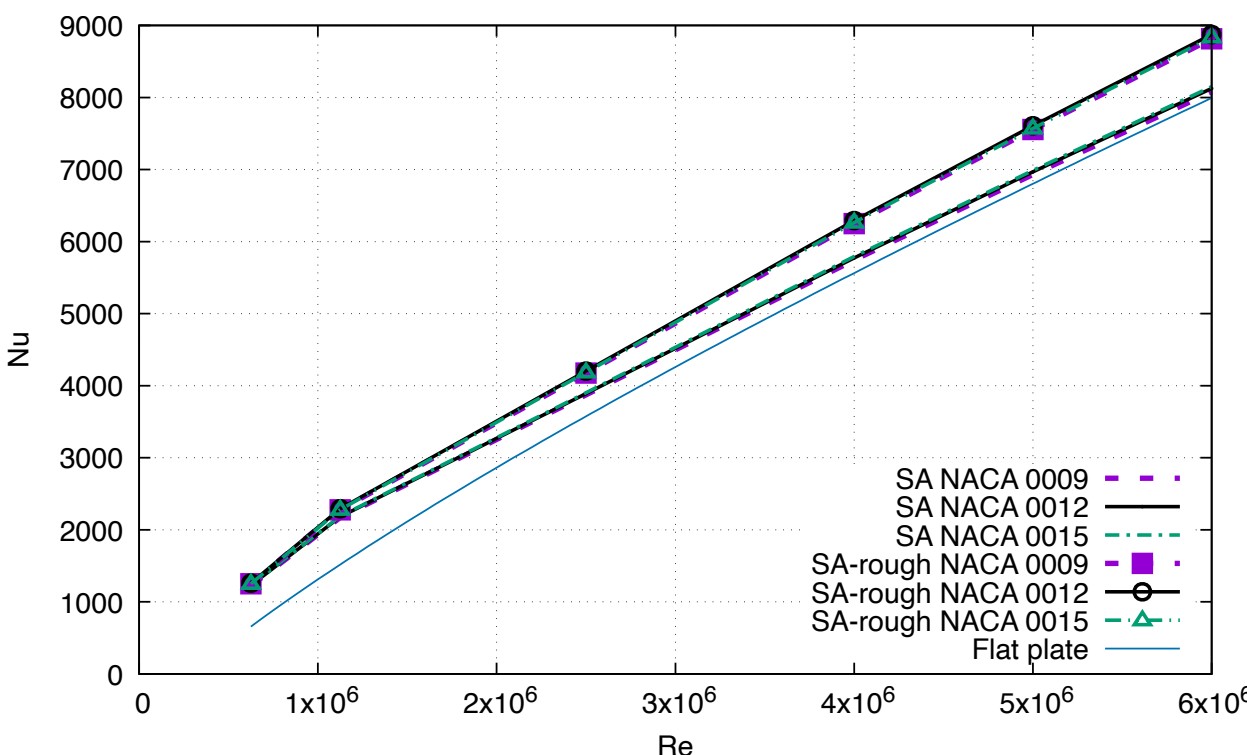

**Figure 8.** $\overline{Nu}$ function of $Re$: comparison between CFD results for the NACA 0009, the NACA 0012, and the NACA 0015 at $\alpha = 0$. Smooth, rough airfoil, and smooth flat plate results.

For symmetric airfoil, $\overline{Nu}$ should be maximum when $\alpha = 0°$. As $c_l$ increases, the stagnation point moves towards the pressure side and the curvilinear distance from the stagnation point to the trailing edge increases on the suction side. The thermal boundary layer becomes thicker on the suction side; thus, the heat transfer is reduced. This is observed on Figure 10a for the three airfoils at $Re = 6 \times 10^6$ for both the smooth and rough surface. $\overline{Nu}$ for the NACA 0009 decreases faster with the lift coefficient than $\overline{Nu}$ for the thicker airfoils. Figure 10b shows that the predicted maximum $c_l$ for the rough airfoil is approximately 0.95 for the NACA 0009, which is below the maximum $c_l \approx 1.1$ for the thicker airfoils. For these airfoils, the flow separation eventually starts at the trailing edge before the maximum lift coefficient is reached. In the separation area, the heat transfer is further reduced. As a consequence, $\overline{Nu}$ for the rough airfoils are lower than for the smooth airfoils, for $c_l > 0.9$ or $c_l > 1.0$ depending on the airfoil thickness. $\overline{Nu}$ shows a stronger dependency on $c_l$ and roughness than on the airfoil thickness.

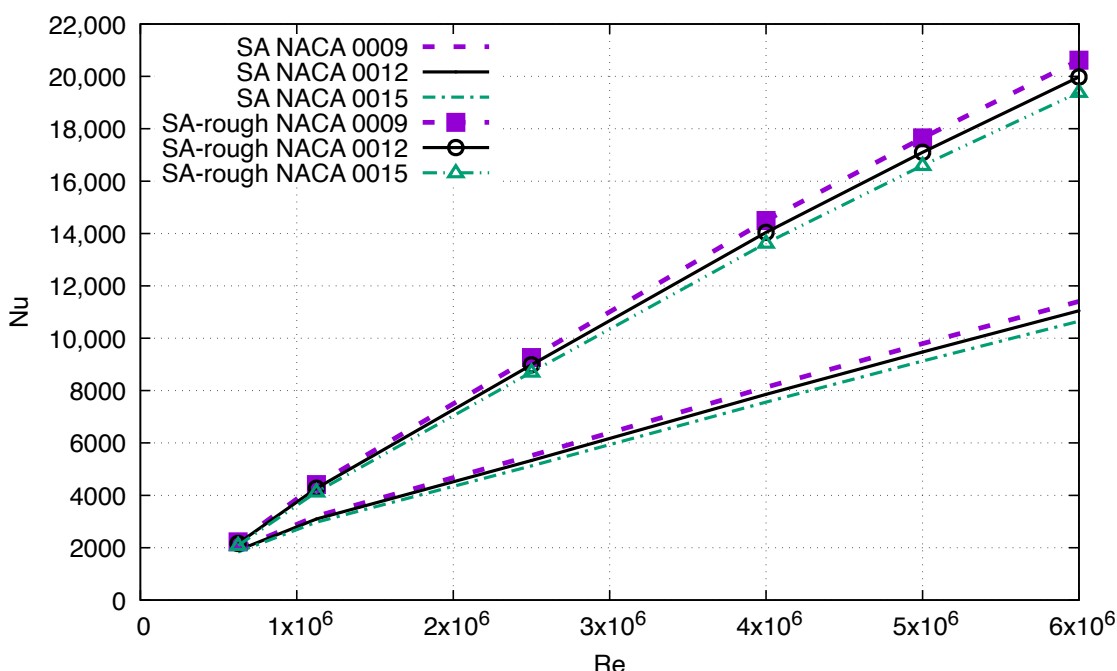

**Figure 9.** $\overline{Nu_r}$ in the leading edge area as a function of *Re*: comparison between the CFD results for the NACA 0009, the NACA 0012, and the NACA 0015 at $\alpha = 0$. Smooth and rough airfoil.

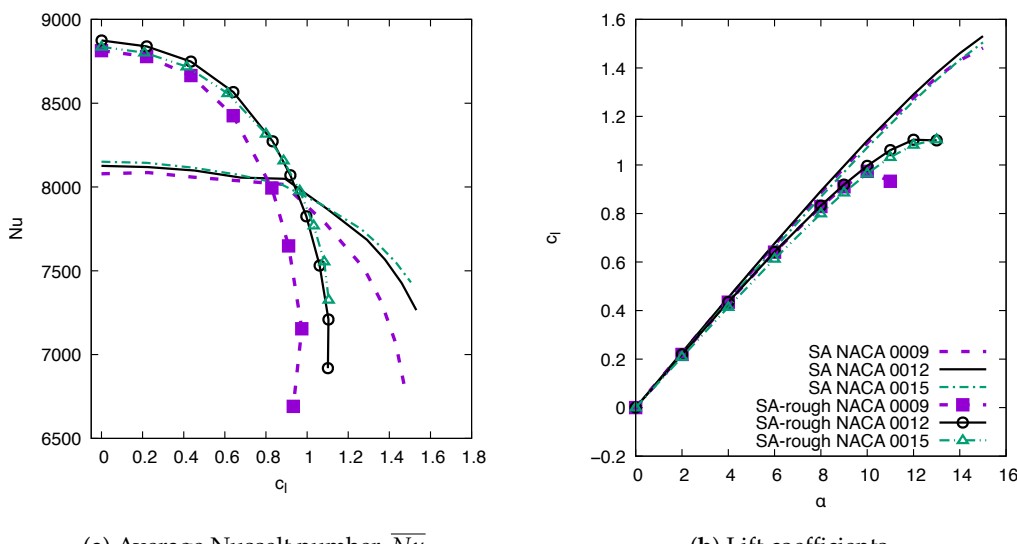

(**a**) Average Nusselt number, $\overline{Nu}$        (**b**) Lift coefficients

**Figure 10.** Comparison between CFD results for the NACA 0009, NACA 0012, and NACA 0015 at $Re = 6 \times 10^6$, smooth and rough leading edge.

### 3.3. Correlations

The average Nusselt numbers for the three smooth airfoils are gathered together to obtain the correlation coefficients. The coefficients are estimated using the Levenberg–Marquardt nonlinear least squares algorithm [51] such that the sum of the squares of the deviations is minimized. The coefficients do not significantly change if the airfoil thicknesses are considered separately. The following correlation fits the data with a coefficient of determination $R^2 = 0.998$

$$\overline{Nu} = (0.0289 Re^{0.81} - 257 c_l^2) Pr^{1/3}. \tag{15}$$

The correlation predictions are compared to the CFD data for the smooth NACA 0012 airfoil on Figure 11. The $\overline{Nu}$ values are plotted at three Reynolds numbers, $Re = 0.625 \times 10^6$,

$2.5 \times 10^6$, and $6 \times 10^6$ as a function of $c_l$. The maximum error on $\overline{Nu}$ is $\pm 250$, and it occurs for the lowest $Re$ number and $c_l > 1.1$.

Similarly, the average Nusselt numbers for the rough airfoils are gathered together. Unlike the smooth correlation Equation (15), coefficient $C$ in front of $c_l^2$ must increases linearly with $Re$ to obtain a coefficient of determination $R^2 = 0.998$

$$\overline{Nu} = (0.0162Re^{0.85} - (2.23 \times 10^{-4}Re)c_l^2)Pr^{1/3}. \tag{16}$$

For the rough airfoils, $\overline{Nu}$ decreases faster with the lift coefficient at a higher $Re$, as shown in Figure 12. At $Re = 1.25 \times 10^6$, the rate of decrease with $c_l^2$ is $C = 279$, close to the $C = 257$ value in the Equation (15). For $Re > 1.25 \times 10^6$, $\overline{Nu}$ is more sensitive to the lift coefficient for the rough airfoils than for the smooth airfoils.

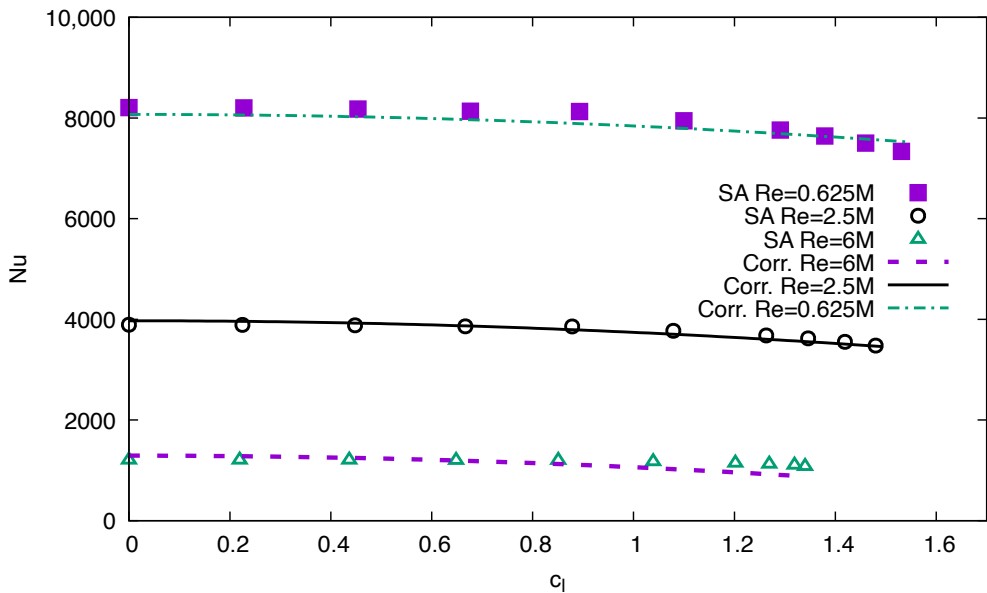

**Figure 11.** $\overline{Nu}$ as a function of $c_l$: comparison between CFD results and correlation for the smooth NACA 0012 airfoil.

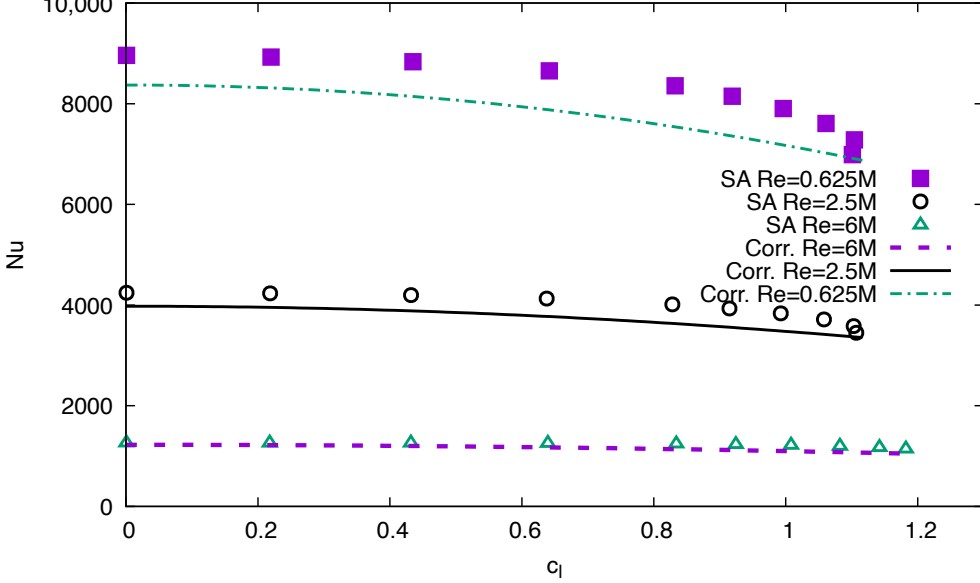

**Figure 12.** $\overline{Nu}$ as a function of $c_l$: comparison between CFD results and correlation results for the rough NACA 0012 airfoil.

## 4. Discussion

This work proposes two correlations to predict the average heat transfer over an airfoil. The correlations are the results of a nonlinear curve fit of a CFD database. The RANS equations and the SA turbulence model poorly predict large flow separations. Hence, the database focuses on fully turbulent flow around the airfoil before the stall angle. The database assumes that the average heat transfer depends on four parameters, $\overline{Nu}$ $(Re, \alpha, h_s, t/c)$. The most important parameter is $Re$. The analysis shows that the correlation coefficients are mostly insensitive to $t/c$ in the studied range, $0.09 \leq t/c \leq 0.15$. To the best of our knowledge, this is the first time the sensitivity of the heat transfer to $t/c$ has been investigated. The average discrepancy between the $\overline{Nu}$ values determined from CFD simulations and the values calculated with the correlation is 3%, with discrepancies reaching $\approx 20\%$ close to the maximum $c_l$ for the lowest $Re$.

The originality of the proposed correlations comes from the fact that they cover Reynolds numbers up to $6 \times 10^6$, relate the average heat transfer coefficient to the lift coefficient, and take into account the surface roughness. Literature correlations for airfoil in free stream flow are summarized in Table 1. They are mostly for laminar or transitional flow [36] or low Reynolds number turbulent flows, $Re < 3 \times 10^6$ [15,38]. Some correlations in the related literature correct the heat transfer by a factor proportional to the angle of attack [15,38], but not the lift coefficients. Furthermore, the present study is the first to suggest that the NACA 0009, NACA 0012, and NACA 0015 could use the same correlation if the average is based on the wet surface.

The correlation results from Table 1 are compared against Equation (15) results for a smooth NACA 0012 at $\alpha = 0°$ in Figure 13. The Equation (15) results agree with [15,36] but extend the range of validity to $Re = 6 \times 10^6$. The results of [38] for a NACA 63-421 are noticeably lower, around $\overline{Nu} = 500$ at $Re = 1 \times 10^6$ instead of $\overline{Nu} \approx 2000$. This is in line with their measured value at the stagnation point. At the stagnation point, the local $Nu \approx 900$ for $Re = 1.037 \times 10^6$, and $\alpha = 0°$ (and $c_l > 0$), approximately five times less than that measured for an NACA 0012. As $Re$ increases, the flat plate correlation results get closer to Equation (15).

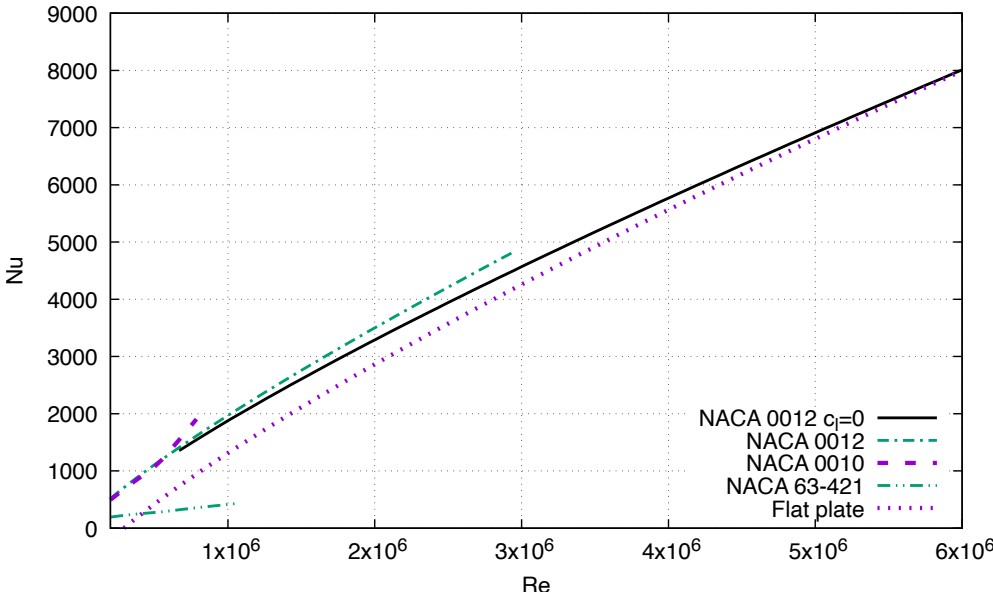

**Figure 13.** $\overline{Nu}$ as a function of $Re$: comparison between the Equation (15) results for the NACA 0012 airfoil at $c_l = 0$ and the correlation results at $\alpha = 0°$ for the NACA 0012 [15], the NACA 0010 [36], the NACA 63-421 [38], and the flat plate $\overline{Nu} = ((0.037Re^{0.8} - 871)Pr^{1/3}$ [23].

**Table 1.** Correlations from the literature.

| Airfoil | $\overline{Nu}$ | $Re$ | $\alpha$ |
|---|---|---|---|
| NACA 0010 [36] | $0.0000422Re^{1.3065}Pr^{1/3}$ | $5 \times 10^5 \le Re < 8.1 \times 10^5$ | $\alpha = 0°$ |
| NACA 0010 [36] | $0.0206Re^{0.8356}Pr^{1/3}$ | $Re < 5 \times 10^5$ | $\alpha = 0°$ |
| NACA 63-421 [38] | $0.0943(0.75 + 0.017\alpha)Re^{0.636}Pr^{1/3}$ | $5 \times 10^5 \le Re < 1.1 \times 10^6$ | $0° < \alpha < 25°$ |
| NACA 63-421 [38] | $2.482(0.75 + 0.013\alpha)Re^{0.389}Pr^{1/3}$ | $Re < 5 \times 10^5$ | $0° < \alpha < 25°$ |
| NACA 0012 [15] | $0.023(1 - 0.389\alpha - 0.678\alpha^2)Re^{0.830}Pr^{1/3}$ | $2 \times 10^5 \le Re < 3.0 \times 10^6$ | $0° < \alpha < 30°$ |

The maximum $\overline{Nu}$ occurs for $\alpha = 0°$. For the smooth airfoils at $Re > 2.5 \times 10^6$, the average heat transfer is only slightly above the one predicted by a correlation for a turbulent flow over a flat plate. As $\alpha$ increases, the lift coefficient increases and $\overline{Nu}$ decreases. For example, a 10% decrease is observed between $\alpha = 0°$ and $15°$ for the smooth NACA 0012 at $Re = 2.5 \times 10^6$. Ref. [38] has experimentally observed this diminution of $\overline{Nu}$ for an NACA 63-421 airfoil up to $\alpha = 15°$. Ref. [15] numerically observed a similar diminution for an NACA 0012 up to $\alpha = 30°$. Their correlations reduce $\overline{Nu}$ by multiplying $Re$ by a coefficient function of $\alpha$, for example

$$\overline{Nu} = A(1 + C_1\alpha + D_1\alpha^2)Re^B Pr^{1/3}.$$

The correlation from [15] set $C_1 = -0.389$, $D_1 = -0.678$, and predicted a 15% reduction in the average heat transfer. In the present work, the suggested correlations use $c_l$ instead of $\alpha$, similarly to most drag coefficient correlations [20].

The authors in [15] suggested a correlation for the average heat transfer over a smooth NACA 0012 based on a CFD database and fully turbulent flows for $0.2 \times 10^6 < Re < 3.0 \times 10^6$. The $Re$ exponent is slightly higher, $B = 0.830$, just outside of the confidence interval for the exponent of Equation (16), $B = 0.81 \pm 0.01$. Experimental heat transfer correlations for different airfoils, the NACA 0010 [36] and the NACA 63-421 [38], also predicted different exponent values. The NACA 0010 results were obtained for laminar and transitional flows, $1.0 \times 10^5 \le Re \le 8.0 \times 10^5$, an angle of attack of $0°$, and constant heat flux. For $Re \ge 5 \times 10^5$,

$$\overline{Nu} = 0.0000422Re^{1.3065}Pr^{1/3}.$$

At $Re = 0.625 \times 10^6$, the correlation predicts $\overline{Nu} = 1390$ for the NACA 0010, whereas Equation (16) predicts $\overline{Nu} = 1290$, only a 7% discrepancy. The NACA 63-421 results were also obtained for transitional flow, with $0.5 \times 10^6 < Re < 1.35 \times 10^6$. The $Re$ exponent is $B = 0.636$. The lower exponent is probably due to the laminar part of the boundary layer flow over the airfoil, since $B = 0.5$ for the average heat transfer over a laminar flat plate [24].

As expected, the leading edge roughness increases the heat transfer [58]. $\overline{Nu}$ increases faster with $Re$ than for the smooth surface, reaching a maximum discrepancy of approximately 10% at $Re = 6 \times 10^6$. Consequently, $Re$ in Equation (16) has a higher exponent, $B = 0.85$, compared to $B = 0.81$ for the smooth Equation (15). If the average only includes the leading area, the average heat transfer almost double at $Re = 6 \times 10^6$. The leading edge roughness also reduces the stall angle and the maximum lift coefficient, as expected. Therefore, the correlation range of validity reduces to $0° < \alpha < 9°$ for NACA 0009. The $\overline{Nu}$ reduction due to the lift coefficient depends on $Re$, in opposition to the smooth airfoil correlation.

No correlation for average heat transfer over a rough airfoil is available in the literature. However, Ref. [59] correlates the $Re$ exponent values with the average roughness height for flow over a flat plate and a maximum correlation constant $B = 0.88$. The determination of the exponent evolution in the case of the leading edge roughness requires CFD calculations with many roughness heights. The results show that the heat transfer is more sensitive to the roughness than to the $c_l$ for $\alpha < 8°$.

These correlations will be useful to extend the use of low-fidelity methods, such as BEMT, or medium fidelity method, such as VLM, to predict the heat required for IPS. Wind turbines, unmanned aerial vehicles [39], or helicopter blades' conceptual design or preliminary design studies frequently use low-to-medium fidelity tools. Often, rotating blades require an ice protection system over the entire airfoil and $\overline{Nu}$ correlations offer a quick estimate of the electrical power needed.

## 5. Conclusions

Two correlations are proposed to predict the average heat transfer for attached flows over an airfoil kept at a constant temperature. The novel form for the correlations relates the average heat transfer to the Reynolds number and the lift coefficient, for fully turbulent flow Reynolds numbers up to $6 \times 10^6$ over smooth and rough surfaces. Equation (15) is valid for a smooth symmetrical airfoil, with $0.09 < t/c < 0.15$, and fully turbulent flow in the range $0.625 \times 10^6 < Re < 6 \times 10^6$. Equation (16) is valid for symmetrical airfoil, with $0.09 < t/c < 0.15$, leading roughness, and fully turbulent flow in the range $0.625 \times 10^6 < Re < 6 \times 10^6$. The leading edge roughness cover the first $0.08c$, the equivalent sand grain roughness is $h_s = 0.001c$ and the roughness height is $h = 0.00458c$. Future works should consider the effects of roughness size and extend around the leading edge.

**Author Contributions:** Conceptualization, F.M.; methodology, S.S. and F.M.; software, F.M.; validation, S.S.; formal analysis, S.S.; investigation, S.S. and F.M.; resources, F.M.; data curation, F.M.; writing—original draft preparation, F.M.; writing—review and editing, F.M.; visualization, S.S.; supervision, F.M.; project administration, F.M.; funding acquisition, F.M. All authors have read and agreed to the published version of the manuscript.

**Funding:** This research received no external funding.

**Institutional Review Board Statement:** Not applicable.

**Informed Consent Statement:** Not applicable.

**Data Availability Statement:** The data presented in this study are openly available in ResearchGate at https://doi.org/10.13140/RG.2.2.12719.20644 (accessed on January 2023).

**Acknowledgments:** This research was enabled in part by support provided by Calcul Québec (https://www.calculquebec.ca/en/) and Digital Research Alliance of Canada (https://alliancecan.ca/) (accessed on February 2023).

**Conflicts of Interest:** The authors declare no conflict of interest. The funders had no role in the design of the study; in the collection, analyses, or interpretation of data; in the writing of the manuscript; or in the decision to publish the results.

## Abbreviations

The following abbreviations are used in this manuscript:

| | |
|---|---|
| BEMT | Blade Element Momentum Theory |
| CFD | Computational Fluid Dynamics |
| CFL | Courant–Friedrichs–Lewy |
| IPS | Ice Protection Systems |
| RANS | Reynolds-averaged Navier–Stokes |
| SA | Spalart–Allmaras One-Equation Model |
| SA-rough | Wall Roughness Correction in Spalart–Allmaras One-Equation Model |
| VLM | Vortex Lattice Method |

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
