# Peer review of "Heat Transfer Correlations for Smooth and Rough Airfoils"

_fluids, doi:10.3390/fluids8020066_

Round 1

Reviewer 1 Report

In this study, the authors performed a numerical analysis on the heat transfer correlations for smooth and rough airfoils. The influences of different structural parameters (NACA type) on heat transfer were identified. The subject of the manuscript falls within the scope of the Fluids.

The subject addressed in this manuscript is worthy of investigation. However, the present manuscript lacks some findings and suffers from shortcomings that I will try to highlight below.

1. The abstract should include a summary of the study's findings.

2. The novelty/originality should be justified by highlighting that the manuscript contains sufficient contributions to the new body of knowledge. The knowledge gap needs to be addressed in Introduction, Results and Discussion, and Conclusion sections.

3. The author needs to discussion more studies that focus on the objective of the research through comparison and evaluation the determinants of those studies and their results, which gives the reader a clear visual of the gap that those studies have not address and covered by this study.

4. The ideas/criteria of choosing the structural parameters of the NACA with and without leading roughness should be given?  

5. In fact, the airfoil is arranged in a vortex generator (VG) device. The aspect of utilization of the VG devices are reported by many investigators. For further enhancement of the literature review, I recommend introducing the following references:

- International Journal of Thermal Sciences 140 (2019) 480–490.

- Journal of Turbulence 23 (2019) pp. 515-547.

- Cogent Engineering 9(1) (2022) 2050021.

- International Journal of Heat and Mass Transfer 201P1 (2023) 123638.

6. Key assumptions and their implications could have been elaborated

7. The author needs to show which model used and why?

8. The flow direction should be added in Fig. 1.

9. When the temperature increases, the properties are changed, for example, viscosity, density, Prandtl number, etc. Please provide details about fluid properties and how is it calculated?

10. Geometric structure of the model is not clear. What is rough surface in Fig. 3. Many parameters are not shown in the Fig. 3 such as material, shape, size, dimensions and etc.

11. Fig. 4(b): The symbol and meaning are missing.

12. Please check the quality of all figure.

13. The author should explain more clearly how to select the optimum solutions related to those indicators.

14. Abstract and conclusion must be modified to show the manuscript novelty and its achievement clearer.

15. In general, for forced convection heat transfer, Nu = function (Re, Pr). Considering all Newtonian fluids, the data may be represented by an algebraic expression of the form Nu = C Re^m Pr^n. The values of C, m, and n are often independent of the nature of the fluid. Thus, for different fluids like air, water and oils flowing through a surface of different sizes and at different velocities, all the data can be collapsed to a single line by plotting the results in terms of Nu/Pr^n as ordinate and Re as abscissa in the log-log plot. Eqs. (17), (18) and (19): why Prandtl number (Pr) is not a function of Nu?

Author Response

Thank you for taking the time to review this manuscript and for your valuable comments.

The corrections are in blue in the text.

  1. The abstract should include a summary of the study's findings.

Answer: We have added the following sentences at the end of the abstract

 For zero lift coefficient, the average Nusselt number is maximum.  It increases with Re0.636 for the smooth surface and with Re0.85 for the rough surface.  As the lift increases, the average Nusselt is reduced by values proportional to the square of the lift coefficient for the smooth surface, while it is reduced by values proportional to Re and the square of the lift coefficient for the rough surface. 

  1. The novelty/originality should be justified by highlighting that the manuscript contains sufficient contributions to the new body of knowledge. The knowledge gap needs to be addressed in Introduction, Results and Discussion, and Conclusion sections.

 Answer: We have added the following sentences before the objectives in the introduction

The design of IPS requires the heat loss by the airfoil to the cold airflow. The BEMT and the unsteady VLM enable quick prediction of the aerodynamic forces and the average heat transfer if they are coupled with correlations to include viscous effects.  However, correlations for average heat transfer are only available for three airfoils with smooth surfaces at Re<3x106. The present study addresses partially this gap by studying the effects of the airfoil thickness, the lift coefficient and the roughness on symmetric airfoil for Re<6x106.

Answer: At the start of the discussion, we have added

The originality of the proposed correlations comes from the fact that they cover Reynolds numbers up to 6x106, relate the average heat transfer coefficient to the lift coefficient, and take into account the surface roughness. Literature correlations for airfoil in free stream flow are mostly for laminar or transitional flow\cite{Benissan-2012} or low Reynolds number turbulent flows, Re<3x106\cite{Wang-2008,Sammad-2020a}.  Some literature correlations correct the heat transfer by a factor proportional to the angle of attack\cite{Wang-2008,Sammad-2020a}, but not the lift coefficients.  Also, the present study is the first to suggest that the NACA 0009, NACA 0012, and NACA 0015 could use the same correlation if average is based on the wet surface.

Answer: At the start of the conclusion

The novel form for the correlations relates the average heat transfer to the Reynolds number and the lift coefficient, for fully turbulent flow Reynolds numbers up to 6x106 over smooth and rough surfaces.

  1. 3. The author needs to discussion more studies that focus on the objective of the research through comparison and evaluation the determinants of those studies and their results, which gives the reader a clear visual of the gap that those studies have not address and covered by this study.

Answer: When possible, we have included in the discussion comparison with previous studies about airfoil in turbulent free stream flow. The table 1 includes selected correlations from literature. Direct comparison of the correlation constants is seldom possible because previous studies use lower Reynolds numbers, different geometries, and different form of the correlations.  No study results are available for average heat transfer over rough airfoil.

  1. The ideas/criteria of choosing the structural parameters of the NACA with and without leading roughness should be given?

Answer: see number 10.

  1. In fact, the airfoil is arranged in a vortex generator (VG) device. The aspect of utilization of the VG devices are reported by many investigators. For further enhancement of the literature review, I recommend introducing the following references:

- International Journal of Thermal Sciences 140 (2019) 480–490.

- Journal of Turbulence 23 (2019) pp. 515-547.

- Cogent Engineering 9(1) (2022) 2050021.

- International Journal of Heat and Mass Transfer 201P1 (2023) 123638.

Answer: We have added in the introduction:

The literature suggests heat transfer correlations for heat exchanger analysis. In particular, recent studies use vortex generators, geometries that share some analogy with wings, to enhance heat transfer in tubes.  The Dittus-Boelter equation,  Nu=0.023 Re0.8 Pr0.4, is modified to consider the geometry, the angle of attack, and the spacing of vortex generators. The correlation coefficients are obtained using the least-square regression method with experimental data \cite{Jayranaiwachira-2023,Zhai-2019}.  Heat transfer enhancement by vortex generators is also numerically studied using CFD\cite{Sharma-2022} data.  Instead of correlations, artificial neural network models are used for performance prediction and optimization of complex heat exchanger geometries\cite{Turgut-2022}.

  1. Key assumptions and their implications could have been elaborated

Answer: We have added in the text, at the beginning of the Materials and methods section:

In the context of rotor aerodynamics, the work assumes that each blade section acts as a quasi-2D airfoil to produce aerodynamic forces and heat transfer, thus 3D effects such as wing tip vortices are not included.  The flow of air is compressible and fully turbulent, no laminar boundary layer region exists on airfoil.  The heat transfer by radiation is neglected. The conduction and the thermal inertia in the airfoil skin are negligible as the temperature is constant in both space and time.

  1. The author needs to show which model used and why?

Answer: As mentioned in the section 2 around line 164, we are solving the compressible RANS equations to model the airflow around the airfoil with the Spalart Allmaras turbulent model.  The air is compressible. We have added below equation 2:

The SA model is commonly used in aeronautics to model attached flow as it gives satisfactory predictions of lift and drag\cite{Blazek-2015}.

  1. The flow direction should be added in Fig. 1.

 Answer: We have added the flow direction.

  1. When the temperature increases, the properties are changed, for example, viscosity, density, Prandtl number, etc. Please provide details about fluid properties and how is it calculated?

Answer: Equation 1 and 2 provide details about viscosity and conductivity dependence with temperature. As mentioned before equation 1, the air is assumed to be a perfect gas. The perfect gas law links the pressure, density and temperature.

  1. Geometric structure of the model is not clear. What is rough surface in Fig. 3. Many parameters are not shown in the Fig. 3 such as material, shape, size, dimensions and etc.

Answer: For the CFD simulations, the geometric structure is not needed because the temperature is assumed to be constant in space and time in the model.  The airfoil is replaced by boundary conditions: no slip boundary conditions (velocity=0) and constant temperature. The rough surface is replaced by an equivalent sand grain roughness and a roughness height. These two parameters change the turbulent viscosity close to the wall and therefore the shear stress and the heat flux. A description of the roughness is added:

For the rough leading edge, the standard leading-edge roughness consists of carborundum grains applied to the surface of the model\cite{Abbott-1959}.  In \cite{Abbott-1959},0.279 mm carborundum grains are applied to a c=0.6096 m airfoil.  The equivalent sand grain roughness is h_s/c=0.001 and the roughness height is h/c=0.00458, corresponding to the small ice accretions between deicing cycles.

  1. Fig. 4(b): The symbol and meaning are missing.

 Answer: We have added in the caption: Legend applies to a) and b)

  1. Please check the quality of all figure.

 Done

  1. The author should explain more clearly how to select the optimum solutions related to those indicators.

Answer: We have added, at the start of the subsection Correlations:

The coefficients are estimated using the Levenberg-Marquardt nonlinear least squares algorithm\cite{Seber-2003}  such that the sum of the squares of the deviations is minimized. 

  1. Abstract and conclusion must be modified to show the manuscript novelty and its achievement clearer.

 See answer to question 1 and 2.

  1. In general, for forced convection heat transfer, Nu = function (Re, Pr). Considering all Newtonian fluids, the data may be represented by an algebraic expression of the form Nu = C Re^m Pr^n. The values of C, m, and n are often independent of the nature of the fluid. Thus, for different fluids like air, water and oils flowing through a surface of different sizes and at different velocities, all the data can be collapsed to a single line by plotting the results in terms of Nu/Pr^n as ordinate and Re as abscissa in the log-log plot. Eqs. (17), (18) and (19): why Prandtl number (Pr) is not a function of Nu?

Answer: We have added the Prandtl number to the correlations.  However, in this paper we study only air with a constant value Pr=0.72. See comment below equation 13:

The Pr dependency is assumed for compatibility with previous correlations.

Reviewer 2 Report

Numerical simulations of the fluid dynamics and heat transfer of air past smooth and rough symmetric airfoils are carried out to provide correlations for the Nusselt number, which enables computing easily the average heat transfer coefficient. Such a quantity is useful in designing ice protection systems: an accurate evaluation not only increases safety but also enables energy saving in the thermal melting system, usually relying upon electric power dissipation.

The open literature provides some experimental correlations developed for smooth airfoils, which are, however, limited in the range of operating conditions and airfoil profiles. No correlation seems available for rough airfoils, which should be better representative of an accreted ice layer.

The numerical simulation is set up well and an accurate discussion of the mesh sensitivity is provided. Moreover, validation of the approach is carefully addressed up to the stall angle by means of the comparison with available experimental data for the fluid dynamic parameters (drag and lift coefficients). Based on the scarce availability of data, validation of the heat transfer simulations is more limited than the fluid dynamic one, but the results are in the expected range according to previous, though less complete, works.

The discussion of the results is very detailed and rigorous. However, to highlight the contribution of the present work and remark the originality and value added, additional plots should be added in Section 4 to compare the developed correlations with the existing ones and to justify the effort of a detailed analysis in a very narrow field of application.

In the annotated draft attached to this review, various suggestions and corrections of misprints are also provided.

Though limited in scope and application, the paper is suited to the journal and very well written, thus I am in favor of publication with minor revisions.

Author Response

Reviewer 2

Thank you for taking the time to review this manuscript and for your valuable comments.

The corrections are in red in the text.

  1. The discussion of the results is very detailed and rigorous. However, to highlight the contribution of the present work and remark the originality and value added, additional plots should be added in Section 4 to compare the developed correlations with the existing ones and to justify the effort of a detailed analysis in a very narrow field of application.

Answer: we have added Figure 13

Round 2

Reviewer 1 Report

The authors have addressed all the comments satisfactorily and the manuscript is now recommended for publication.